# History information emerges in the cortex during learning

**Odeya Marmor[1]\*, Yael Pollak[1], Chen Doron[1], Fritjof Helmchen[2,3], Ariel Gilad[1]\***

[1]Department of Medical Neurobiology, Faculty of Medicine, The Institute for Medical Research Israel-Canada (IMRIC), The Hebrew University of Jerusalem, Jerusalem, Israel; [2]Brain Research Institute, University of Zurich, Zurich, Switzerland; [3]Neuroscience Center Zurich, Zurich, Switzerland

**Abstract** We learn from our experience but the underlying neuronal mechanisms incorporating past information to facilitate learning is relatively unknown. Specifically, which cortical areas encode history-related information and how is this information modulated across learning? To study the relationship between history and learning, we continuously imaged cortex-wide calcium dynamics as mice learn to use their whiskers to discriminate between two different textures. We mainly focused on comparing the same trial type with different trial history, that is, a different preceding trial. We found trial history information in barrel cortex (BC) during stimulus presentation. Importantly, trial history in BC emerged only as the mouse learned the task. Next, we also found learning-dependent trial history information in rostrolateral (RL) association cortex that emerges before stimulus presentation, preceding activity in BC. Trial history was also encoded in other cortical areas and was not related to differences in body movements. Interestingly, a binary classifier could discriminate trial history at the single trial level just as well as current information both in BC and RL. These findings suggest that past experience emerges in the cortex around the time of learning, starting from higher-order association area RL and propagating down (i.e., top-down projection) to lower-order BC where it can be integrated with incoming sensory information. This integration between the past and present may facilitate learning.

## Editor's evaluation

This is important work analyzing the trial-by-trial progression of learning, and how the outcome of one trial influences cortex-wide neural responses on the next trial. The strength of the evidence is compelling, with control experiments provided to rule out potential confounds of hemodynamic effects and extensive analyses provided to address the challenging issue of potential behavioral changes induced by the previous trial.

**\*For correspondence:**
odeya.marmor@gmail.com (OM);
ariel.gilad@mail.huji.ac.il (AG)

**Competing interest:** The authors declare that no competing interests exist.

## Introduction

Learning is a process of acquiring new knowledge required for appropriate behavior and is highly dependent on our previous experience. Our brain integrates incoming sensory information with history information of previous stimuli to form a knowledgeable association of the current stimulus. Although the strong link between history (i.e., past experience) and learning, the underlying cortex-wide dynamics are relatively unknown, partially because most previous studies separately focus either on learning or history (*Hattori et al., 2019*). Learning-related neuronal dynamics are broadly observed across the whole cortex, including primary sensory and motor areas (*Blake et al., 2002*; *Chen et al., 2015*; *Gilad and Helmchen, 2020*; *Jurjut et al., 2017*; *Komiyama et al., 2010*; *Li et al., 2008*; *Poort et al., 2015*; *Makino and Komiyama, 2015*; *Yan et al., 2014*), higher-order association areas (*Driscoll*

*et al., 2017*; *Gilad and Helmchen, 2020*), and prefrontal cortex (*Le Merre et al., 2018*; *Pasupathy and Miller, 2005*). But do these areas that participate in the learning process also carry trial history information?

Encoding of information carried on from the previous trial (i.e., trial history) has been reported mainly in higher-order cortical areas such as the posterior parietal cortex (PPC) (*Akrami et al., 2018*; *Harvey et al., 2012*; *Hwang et al., 2017*; *Morcos and Harvey, 2016*; *Scott et al., 2017*; *Suzuki et al., 2022*), retrosplenial cortex (*Hattori et al., 2019*; *Vann et al., 2009*), and prefrontal cortex (*Banerjee et al., 2020*; *Johnson et al., 2016*; *Kawai et al., 2015*; *Scott et al., 2017*; *Sul et al., 2010*; *Tsutsui et al., 2016*), and to a smaller extent in lower-order primary sensory areas such as BC (*Banerjee et al., 2020*; *Chéreau et al., 2020*; *Rodgers et al., 2021*). It is still unknown on how different cortical areas encode trial history with regard to learning. In other words, does trial history encoding in the cortex change as a function of learning? Another important aspect of the history learning is the temporal relationship between trial history encoding and the current stimulus. For example, does trial history emerge in cortex before the current incoming stimulus, or maybe both past and present information emerge simultaneously in a certain cortical area? From the temporal aspect, optogenetic silencing of PPC area during the inter-trial interval affected performance, highlighting that higher-order cortical areas may maintain history information before the incoming current stimulus (*Akrami et al., 2018*; *Hwang et al., 2017*).

To study the history-learning relationship, we use wide-field cortical imaging of mice learning to discriminate between two textures and focus on the cortex-wide dynamics of trial history. In a previous study using the same dataset, we showed that mice learning a whisker-based texture discrimination task, increase activity in task-related areas (e.g., BC and rostrolateral association cortex [RL]) as they become experts (*Gilad and Helmchen, 2020*). RL is part of the PPC and is located within the cluster of higher-order association areas surrounding V1. RL plays pivotal roles in cross-modal sensory integration, learning, and history, but the relationship between history and learning in RL is relatively unknown (*Akrami et al., 2018*; *Driscoll et al., 2017*; *Hattori et al., 2019*; *Hwang et al., 2017*; *Khodagholy et al., 2017*; *Morcos and Harvey, 2016*; *Save and Poucet, 2009*). By classifying trials according to the preceding trial, we were able to detect trial history information that emerges only as the mouse gains expertise. Specifically, trial history emerges in RL, before stimulus presentation and then is transferred to BC during stimulus presentation, which may aid in learning the rewarded stimulus.

## Results

In this study, we investigate trial history dynamics across the whole dorsal cortex and its emergence during learning in transgenic mice expressing a calcium indicator (GCaMP6f) in L2/3 excitatory neurons (*n* = 7 mice). This dataset is identical to the one published in *Gilad and Helmchen, 2020* where we focused only on learning dynamics. Using wide-field calcium imaging through the intact skull (*Gallero-Salas et al., 2021*; *Gilad et al., 2018*; *Gilad and Helmchen, 2020*; *Vanni and Murphy, 2014*), we chronically measured large-scale neocortical L2/3 activity in the contralateral hemisphere as mice learned a go/no-go whisker-dependent texture discrimination task (*Gilad and Helmchen, 2020*). Whisker movements and body movements were video monitored and synchronized to the calcium imaging data (Materials and methods). To map the dorsal cortex, we functionally mapped sensory areas for each mouse during anesthesia (see Materials and methods). Based on these maps (and skull coordinates) we registered all images to the 2D topview Allen reference atlas (*Oh et al., 2014*) and defined 25 areas of interest, further divided into four groups (*Figure 1c*; *Gilad and Helmchen, 2020*).

Mice were trained on a head-fixed, whisker-based go/no-go texture discrimination task (*Chen et al., 2013*; *Gilad and Helmchen, 2020*; *Figure 1a*; Materials and methods). Each trial started with an auditory cue (stimulus cue), signaling the approach of either two types of sandpapers (grit size P100: rough texture; P1200: smooth texture; 3M) to the mouse's whiskers as 'go' or 'no-go' textures. The texture stayed in touch with the whiskers for 2 s, and then it was moved out after which an additional auditory cue (response cue) signaled the start of a 2-s response period (*Figure 1b*) followed by a 6-s break until the next trial auditory cue. Five mice were trained to lick for the P100 and two mice were trained to lick for the P1200 texture. Mice were rewarded in 'Hit' trials for correctly licking after the go texture and punished with white noise for incorrectly licking for the no-go texture ('false alarm' trials, FA). Mice were neither rewarded nor punished when they withheld licking for the go and no-go textures ('Miss' and 'correct-rejection', CR, trials, respectively). We defined two time windows within

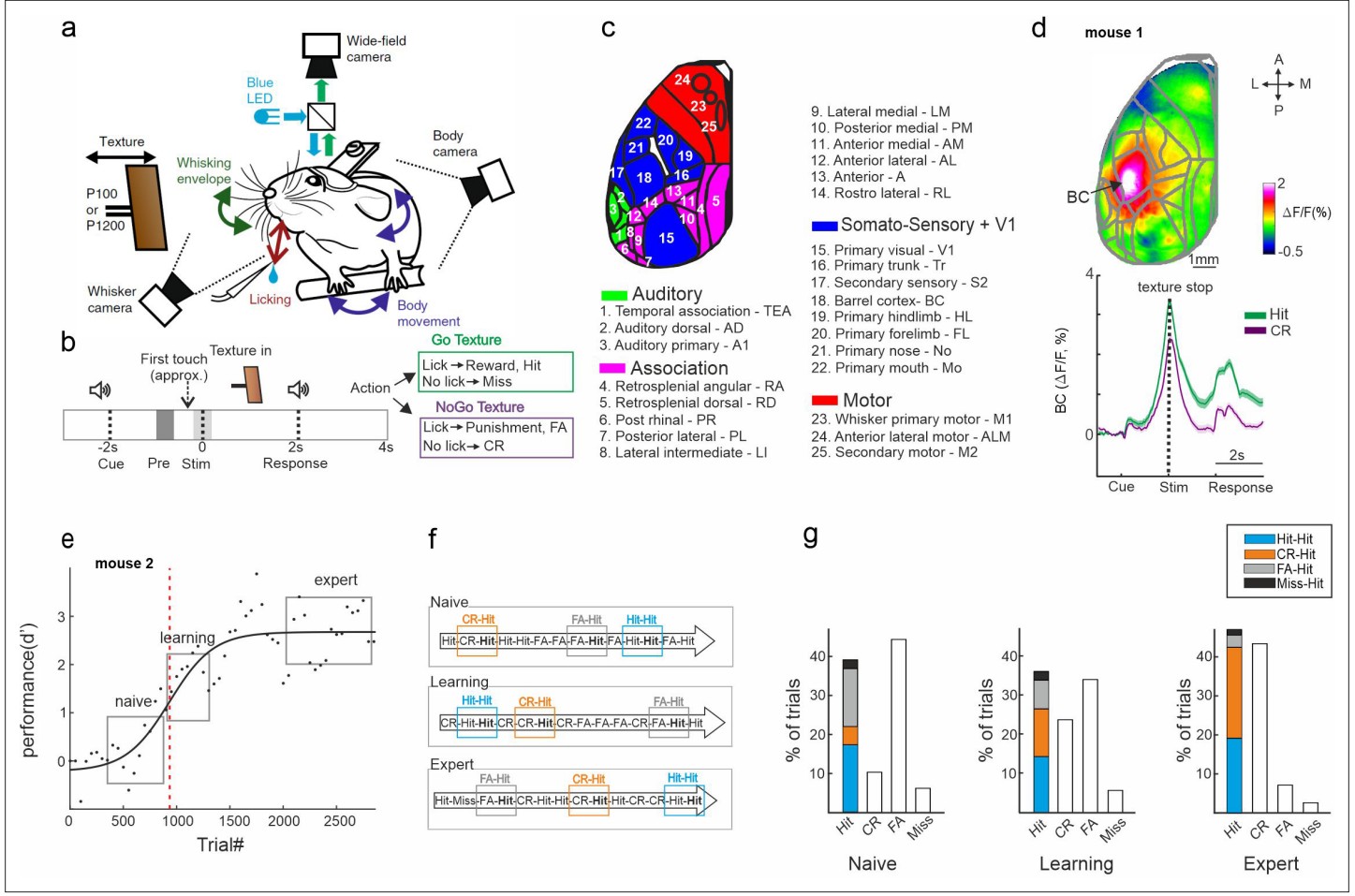

**Figure 1.** Trial types based on history. (**a**) Behavioral setup for head-fixed texture discrimination with simultaneous wide-field calcium imaging and video monitoring of whisker motion and body movement. (**b**) Trial structure and possible trial outcomes. pre and stim periods are marked in gray and light gray colors, respectively. (**c**) Twenty-five cortical areas used in this study grouped into auditory areas (green), association areas (pink), somatosensory + V1 areas (blue), and motor areas (red). (**d**) Top: Example mean activation map (averaged during the stim period) for the Hit condition. BC – barrel cortex. Color denotes normalized fluorescence. Bottom: Time course of activity in BC for Hit (green) and correct rejection (CR; purple). Error bars are mean ± standard error of the mean (SEM) across trials (*n* = 376 and 333 for Hit and CR, respectively). (**e**) Example of a learning curve (*d'* as a function of trial number) of one mouse, fitted with a sigmoid function (solid black line). Red dashed vertical line indicates the learning threshold. gray rectangles mark the naive, learning, and expert phases. (**f**) Schematic diagram of the different trial types for a Hit trial preceded by a different trial (i.e., history): Hit-Hit (blue), CR-Hit (orange), and FA-Hit (gray). (**g**) Probability of the different trial types along with the distribution of history for the Hit trial during the naïve, learning, and expert phases (averaged across seven mice).

The online version of this article includes the following figure supplement(s) for figure 1:

**Figure supplement 1.** Learning curves of all seven mice.

**Figure supplement 2.** Individual learning curves based on trial history.

**Figure supplement 3.** Probability of a false alarm (FA) based on a different preceding trial type: Hit (pink), correct rejection (CR; green), or Miss (purple).

the trial structure: the 'pre period' when the texture approaches the whiskers (−1 to −0.6 s relative to the texture stop; mainly before the first whisker-texture touch); and the 'stim period' during texture touch (−0.2 to 0.2 s relative to texture stop; *Figure 1b*).

The performance of all mice increased with training (5–11 days; ~500 trials/day) and eventually reached high discrimination levels (quantified by *d'*; *Figure 1—figure supplement 1*; *Gilad et al., 2018*; Materials and methods). We defined the 'learning threshold' of reaching expert level for each mouse by crossing the inflection point of the sigmoid fit for the learning curve (in units of 'trial number'; *Figure 1e*, *Figure 1—figure supplement 1*). The fastest learning mouse reached threshold in slightly less than thousand trials whereas mouse #4 took substantially longer (*Figure 1—figure supplement 1*). In addition, we defined a naive (first day of recording), learning (day of crossing the learning

threshold; second or third day), and expert (last recording day) phases for each mouse (*Figure 1e*). All mice, after gaining expertise, showed strong activation in the BC (*Figure 1d*, upper panel). This activation was during stimulus representation, stronger in Hit trials compared to CR trials (*Figure 1d*, lower panel), not dependent on the texture type (i.e., if the hit was P100 or P1200).

Here, we focus on the trial history content for each trial type. We subgrouped all the Hit trials (i.e., the current trial type) based on the previous trial type (i.e., trial history): CR ('CR-Hit'; $n = 423 \pm 74$, mean ± standard error of the mean [SEM]), Hit ('Hit-Hit'; $n = 585 \pm 42$), FA ('FA-Hit'; $n = 217 \pm 24$), or Miss ('Miss-Hit'; $n = 55 \pm 24$; *Figure 1f, g*). 'Miss-Hit' were not analyzed due to a small number of trials. Our main analysis will compare 'CR-Hit' (orange) and 'Hit-Hit' (blue) trial pairs, since they are present in large numbers during all phases in each mouse separately (*Figure 1g*; but see *Figure 2—figure supplement 7* for a comparison of other trial pairs). We note that learning curves that are calculated separately for each pair (i.e., either a preceding Hit or CR trial) were not significantly different (*Figure 1—figure supplement 2*). We further note FA probabilities did not significantly differ based on the preceding trial type (*Figure 1—figure supplement 3*). In addition, the lick reaction time (but not the lick rate) between Hit-Hit and CR-Hit were significantly different ($p < 0.05$; Wilcoxon signed-rank test). We emphasize that in this comparison, the current trial type is identical (i.e., Hit) whereas only the pervious trial (i.e., the history, CR, or Hit) differed, therefore eliminating activity differences due to the current stimulus.

## Trial history in BC emerges during learning

First, we focused on trial history encoding in BC, specifically during the stim period. BC displayed higher activity during CR-Hit compared to Hit-Hit only during learning and expert phases, but not during the naive period (*Figure 2a*, *Figure 2—figure supplement 1*). This difference was significant during the stim period in learning and expert phases across mice (*Figure 2b*; two-way analysis of variance [ANOVA] with repeated measures; DF(1-6) $F = 51$ $p < 0.001$, DF(2-12) $F = 18$ $p < 0.001$, DF(2-12) $F = 5$ $p < 0.05$ for trial history, learning, and the interaction between trial history and learning; post hoc Tukey analysis $p < 0.05$ for trial history in learning and expert phases; $p > 0.05$ in the naive phase). In addition, a statistical comparison between CR-Hit and Hit-Hit responses within each mouse separately maintained significance for expert (7/7 mice; Mann–Whitney $U$-test $p < 0.05$) and learning (6/7 mice) but not for naive (0/7 mice) (*Figure 2—figure supplement 1*). We further report that responses during the reward period in cortex and specifically in BC went back to baseline 4–5 s after the start of the reward period and 6–8 s before the presentation of the next stimulus (total inter-trial interval ranged between 10 and 12 s). In addition, responses in BC during the reward period were not consistently modulated as a function of learning ($p > 0.05$; Wilcoxon signed-rank test between naive and expert, BC response averaged during the reward period, 2–4 s after stimulus onset; $n = 7$ mice). Taken together, we find that direct responses from the reward period do not affect history-related responses during the next trial.

To control for the possible contamination of non-calcium-related signals such as hemodynamics, we performed a battery of additional experiments (see Materials and methods). First, two mice performed the task (expert) while we excited the cortex with a control light (510 nm, isosbestic wavelength). Correcting the original signal with the control light maintained significant difference in trial history (*Figure 2—figure supplement 2*). Second, we trained an additional three mice on the same task and imaged their cortex using an interleaved protocol of 473 (calcium signal) and 405 (control signal, isosbestic wavelength) nm lights. The corrected signal (473 signal minus the 405 signal; see Materials and methods) maintained a significant trial history difference between CR-Hit and Hit-Hit conditions during learning and expert phases, but not during the naive phase (*Figure 2—figure supplement 3*). Finally, we further performed two photon imaging of single cells in BC and RL during the expert phase and found significant trial history differences, that is, higher response in CR-Hit compared to Hit-Hit in BC (during the stim period) and RL (*Figure 2—figure supplement 4*; during the pre period). Taken together, non-calcium dynamics such as hemodynamics have a minor effect the results, specifically regarding trial history differences.

To check whether this effect is not due to difference in body or whisker movements between the two pair types, we calculated the body movements (1 − frame-to-frame correlation in mouth, forelimb and hindlimb areas) and whisker envelope as a function of time (see Materials and methods). Both body movements and whisker envelope were similar between CR-Hit and Hit-Hit pairs (*Figure 2c*) and

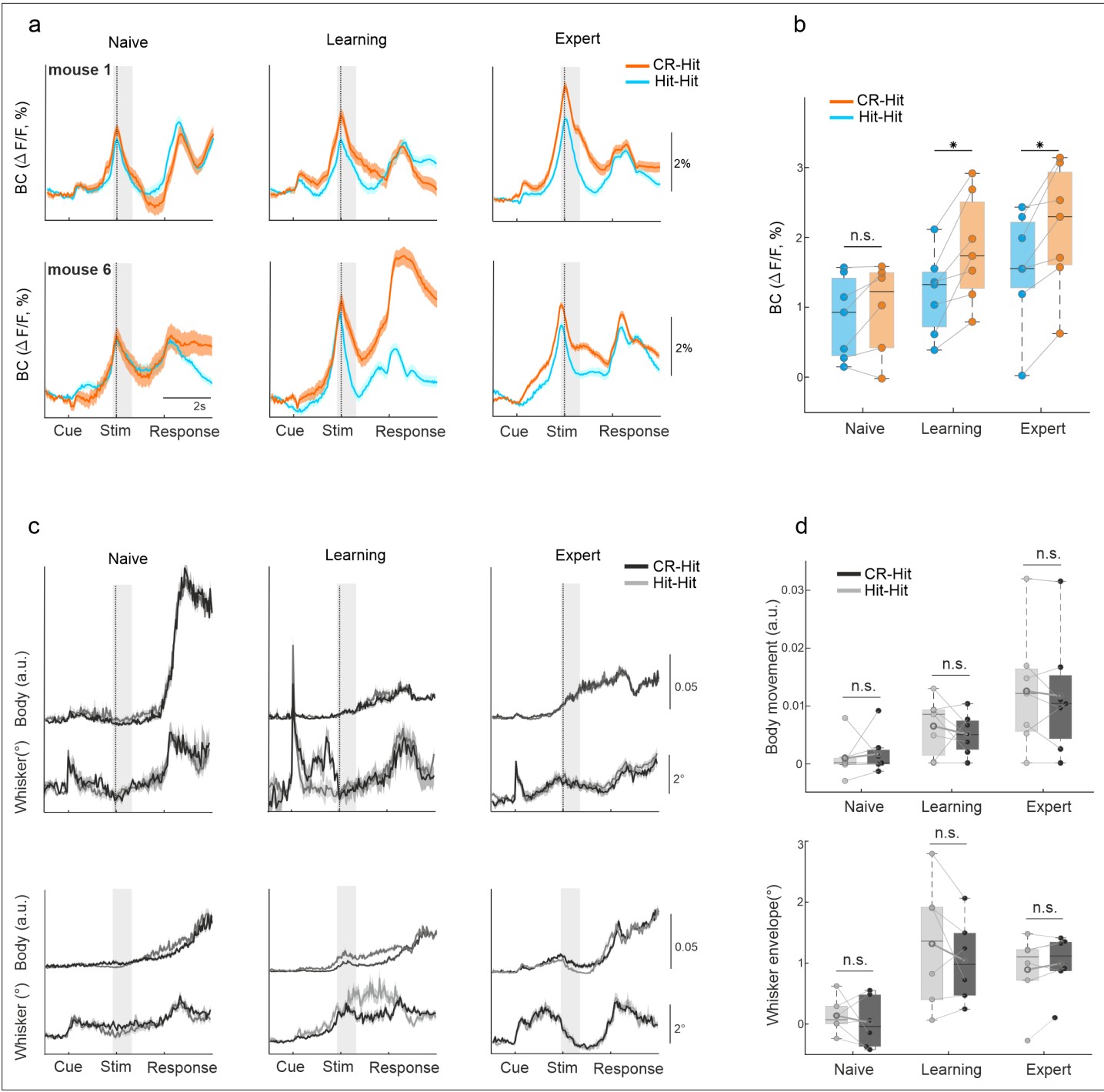

**Figure 2.** History information in barrel cortex (BC). (**a**) Example of average BC response of Hit-Hit (blue) and CR-Hit (orange) from two mice (upper and lower rows) in the naive, learning, and expert phases. Shaded bar depicts the stim period. Error bars are mean ± standard error of the mean (SEM) across trials (mouse 1: $n$ = 86/66, 90/70, and 166/173 Hit-Hit/CR-Hit for naive, learning, and expert phases, respectively; mouse 6: $n$ = 94/80, 86/121, and 99/135). (**b**) Grand average of BC activity during the stim period (−0.2:0.6 ms) for the naive, learning, and expert phases. Error bars are mean ± SEM across mice ($n$ = 7). (**c**) Same as (**a**) but for body and whisker movements in the Hit-Hit (light gray) and CR-Hit (dark gray) trials. (**d**) Same as (**b**) but for body (top) and whisker (bottom) movements. *$p < 0.05$; n.s. – not significant; Wilcoxon signed-rank test.

The online version of this article includes the following figure supplement(s) for figure 2:

**Figure supplement 1.** History information in barrel cortex (BC) for each mouse separately.

**Figure supplement 2.** Correction for hemodynamic contamination.

*Figure 2 continued on next page*

*Figure 2 continued*

**Figure supplement 3.** Correction for hemodynamic signal maintains history information.

**Figure supplement 4.** Two-photon single neurons encode trial history similar to the population signals.

**Figure supplement 5.** Decomposing body movements in Hit-Hit and CR-Hit conditions.

**Figure supplement 6.** Single whisker parameters do not differ between CR-Hit and Hit-Hit.

**Figure supplement 7.** Activity in barrel cortex (BC) for other trial pairs.

there was no significant difference across mice during the stim period for neither naive, learning, or expert phases (*Figure 2d*, p > 0.05; signed-rank test) nor during the pre period (p > 0.05, signed-rank test, data not shown). A within mouse statistical comparison between body or whisker parameters in CR-Hit and Hit-Hit maintained a non-significant difference in expert (1/7 mice were significantly different; Mann–Whitney *U*-test p > 0.05), learning (2/7 mice) and naive (0/7 mice). In addition, we performed a more detailed body and whisker analysis, for example, decomposing the movement to different body parts and obtaining single whisker dynamics. These analyses did not find significant differences in movement parameters between CR-Hit and Hit-Hit conditions (*Figure 2—figure supplement 5* and *Figure 2—figure supplement 6*). These results, along with the fact that the current trial type in both conditions is identical, strongly indicate the presence of trial history information in BC.

We next quantified the emergence of trial history with regard to the different time scales, the trial structure (within seconds) or the learning profile (across days). We first show 2D activity plots in BC for each trial pair (i.e., CR-Hit and Hit-Hit; showing activity of only the Hit trial), where trial time is plotted on the x-axis and trial number across learning time on the y-axis (*Figure 3a*; 100-trial bins regardless of trial pair). Both trial pairs display an increase in activity during the stim period slightly after passing the learning threshold. We defined a history modulation index as the difference in activity for BC between the two pair types (Hit-CR minus Hit-Hit). History modulation increased around the stim period only in learning and expert phases but not in the naive case (*Figure 3b, c*). A significant history modulation was defined as values exceeding mean ± 2 standard deviation (SD) of a trial-shuffled sample distribution (*n* = 1000 iterations) and was performed for each mouse separately (*Figure 3b*). The onset of the history modulation was defined as the first-time frame reaching significant values (red arrows in *Figure 3b*) and was found in BC within the stim period (*Figure 3d*; 0.08 ± 0.28 s, −0.32 ± 0.28 s, median ± SEM relative to texture stop in learning and expert phases, respectively). We note that in the expert phase there is also a small peak exceeding the significance around the cue, indicating history information in BC may be present to some extent before stimulus presentation. Next, we quantified the history modulation in BC during the stim period as a function of the learning profile. History modulation in BC had the steepest increase after mice crossed their learning threshold (*Figure 3e, f*). The onset of the history modulation was defined as the first trial bin exceeding mean ± 2 standard deviation of trial-shuffled sample distribution and was found to occur shortly after the learning threshold, highly correlated with the learning threshold indicating strong relationship between history emergence and learning of each individual mouse (*Figure 3g, h*; 500 ± 221 trials, median ± SEM, *r* = 0.97, p < 0.001, Spearman correlation). Note that our definition of significance is relatively strict and an increase in history information can be observed shortly (i.e., tens of trials) after crossing the learning threshold (*Figure 3e*).

We expanded our trial history analysis also for the other pair types other than CR-Hit and Hit-Hit. For sufficient trial numbers, we focused on the learning phase. First, we compare FA-Hit to Hit-Hit and CR-Hit, that is, the same current trial type but preceded by an error trial (FA). Response in BC for FA-Hit was similar to Hit-Hit and significantly lower compared to CR-Hit (*Figure 2—figure supplement 7*; p < 0.05 signed-rank test). This result highlights that specifically a correct rejection (CR), rather than the stimulus (i.e., texture) type, has a strong history effect. Next, we compared FA-CR, Hit-CR, and CR-CR, that is, similar to the previous comparison differing only in the current trial type (CR instead of Hit). There was no significant difference between the different pairs, indicating that the current trial type, that is, Hit in this case, has a strong effect along with the history of the CR (*Figure 2—figure supplement 7*; p > 0.05, signed-rank test). A comparison of FA-FA, Hit-FA, and CR-FA did not show a significant difference (*Figure 2—figure supplement 7*; p > 0.05, signed-rank test). In general, a preceding CR trial resulted in higher activation independent of the current trial type (i.e., Hit, CR, or

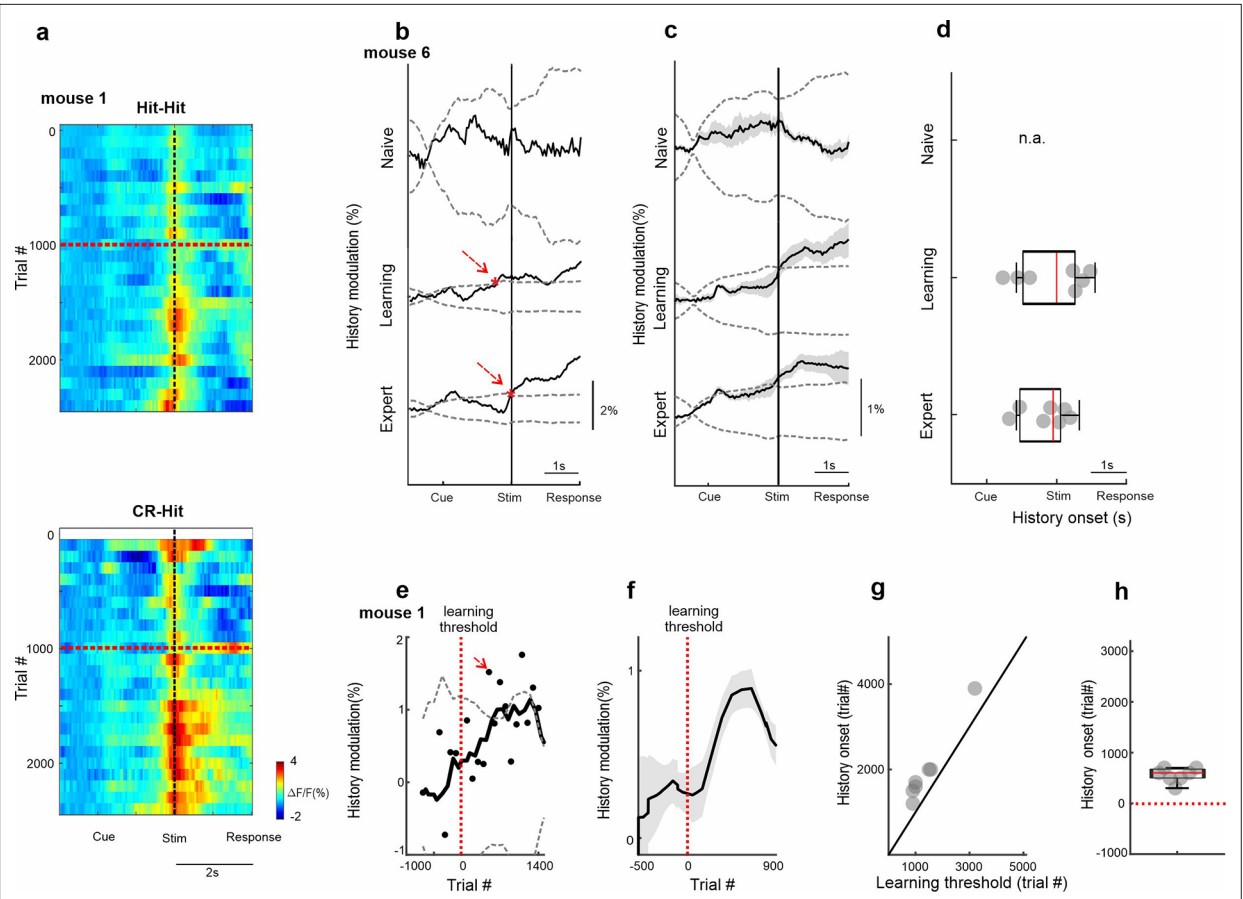

**Figure 3.** Temporal dynamics of history information in barrel cortex (BC). (**a**) 2D plot of BC responses for Hit-Hit (top) and CR-Hit (bottom; trial structure on x-axis; trial number across learning in bins of 100 trials) on the y-axis. Red horizontal dashed line indicates learning threshold. Black dashed vertical line indicates the time of texture stop. (**b**) Example from one mouse of the history modulation (activity in CR-Hit minus activity in Hit-Hit) in BC along the trial structure in the naive, learning, and expert phases. Dashed gray line is the mean ± 2 standard deviation (SD) of the trial-shuffled data (n = 1000 iterations). The first-time frame crossing the shuffle data is defined as the onset and is marked in red. (**c**) Mean history modulation in BC along trial time. Error bars depict mean ± standard error of the mean (SEM) across mice (n = 7). (**d**) Median onset of history modulation. Error bars depict median ± SEM across mice (n = 7). (**e**) Example from one mouse of the history modulation along learning dimension. Dashed gray line is the mean ± 2 SD of the trial-shuffled data (n = 1000 iterations). The first-time frame crossing the shuffle data is defined as the onset for learning and is marked in red. The vertical red dashed line (trial 0) marks the learning threshold. (**f**) Mean history modulation in BC along the learning profile aligned to the learning threshold of each mouse (time 0). Error bars depict mean ± SEM across mice (n = 7). (**g**) Onset of the history modulation for learning as a function of the learning threshold. Each point is one mouse (n = 7). (**h**) Median onset of history modulation relative to the learning threshold. Error bars depict median ± SEM across mice (n = 7).

FA; not significant for CR and FA), indicating that history information is present at the current time independently of incoming sensory information (***Figure 2—figure supplement 7***; compare orange bars to the blue bars). In conclusion, we found that the CR-Hit pair displayed a specific enhancement in BC which is related both to the preceding and current trial type (see Discussion).

Next, we expanded our analysis to the whole dorsal cortex during the stim period. Mean activation maps for both CR-Hit and Hit-Hit pairs (i.e., activity for the current Hit trial whereas only the preceding trial was different) during the stim period displayed a pronounced activation patch in BC during naive, learning, and expert phases (***Figure 4a***). BC activity was higher in CR-Hit compared to Hit-Hit especially during learning and expert phases. The grand average activity for all 25 cortical areas highlights history-dependent information that emerges during learning (***Figure 4b***). We note that other areas, for example, different association areas, also encoded trial history information especially during learning and expert phases. In addition, we present activity difference maps between CR-Hit and Hit-Hit conditions during the stim period (***Figure 4—figure supplement 1a***). These maps clearly show the highest trial history information (i.e., difference in activity) in BC. Taken together, these results

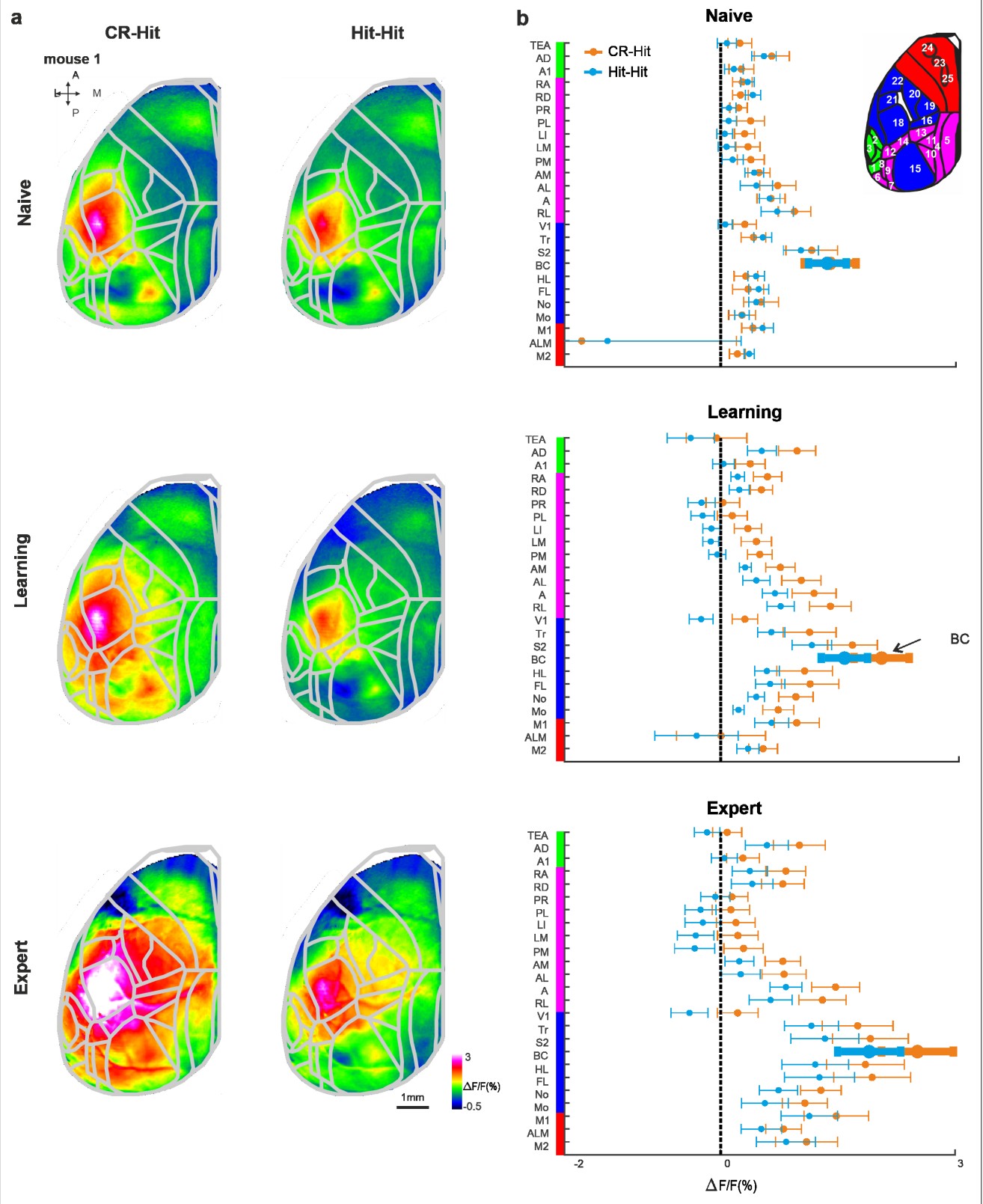

**Figure 4.** Cortex-wide history modulation during the stim period. (**a**) Mean activity maps averaged within the stim period (−0.2 to 0 s relative to texture stop) of CR-Hit (left) Hit-Hit (right) during the naive (top), learning (middle), and expert (bottom) phases. Color bar denotes normalized fluorescence (ΔF/F). 2D top-view atlas is superimposed in gray. (**b**) Grand average neuronal activity during the stim period (−0.2:0.2 s) for Hit-Hit (blue) and CR-Hit

*Figure 4 continued*

(orange) in all 25 areas for the naive (top), learning (middle), and expert (bottom) phases. Error bars depict mean ± standard error of the mean (SEM) across mice (*n* = 7).

The online version of this article includes the following figure supplement(s) for figure 4:

**Figure supplement 1.** Activity difference map during stim and pre periods.

indicate that BC encodes trial history information that emerges during the stim period and just after learning. These results gave us the motivation to examine history-dependent information at time periods before texture touch.

## Trial history in RL before sensation

We next focused our analysis on the pre period, just before texture touch (−1 to −0.6 s before texture stop). Mean activity maps during the pre period highlight activity in association area RL that is present for both CR-Hit and Hit-Hit pairs during the naive, learning, and expert phases (*Figure 5a*; *Gilad and Helmchen, 2020*). RL pre period activity is higher in CR-Hit compared to Hit-Hit mostly during learning and expert phases. In addition, higher RL activity in CR-Hit pair starts even before the pre period, indicating that trial history is not directly related to the current stimulus (*Figure 5b*). The grand average of all 25 cortical areas, highlights the emergence of trial history during learning, especially in RL, but also in other association and sensory areas (*Figure 5c*). In addition, we present activity difference maps between CR-Hit and Hit-Hit conditions during the pre period (*Figure 4—figure supplement 1b*). These maps localize trial history information to RL which also spreads to other adjacent association areas. Moreover, activity patches slightly vary across the different mice which may affect the grand average (averaged across mice) of each area.

RL activity was significantly higher in CR-Hit compared to Hit-Hit trials in the pre period during the expert phase (*Figure 5—figure supplement 1*; signed-rank test, p < 0.05, similar trend for the learning phase but insignificant; not significant for the naive phase). In addition, a statistical comparison between CR-Hit and Hit-Hit responses in RL within each mouse separately maintained significance for expert phase (5/7 mice; Mann–Whitney *U*-test p < 0.05). Body movements and whisker parameters did not significantly differ between CR-Hit and Hit-Hit conditions during the pre period (Similar to the stim period. Across and within mice. p > 0.05; Mann–Whitney *U*-test). The onset of history modulation within the trial structure (as in *Figure 3d*) was earlier in RL compared to BC in both learning (−0.15 ± 0.85 and 0.05 ± 0.86 s, median ± SD in RL and BC, respectively) and expert phases (−0.75 ± 0.53 and −0.1 ± 0.74 s, median ± SD in RL and BC, respectively) but not significantly different (p > 0.05, signed-rank test). The onset for the history modulation with relation to the learning profile in RL (similar to *Figure 3h*; During the pre period) was also earlier than BC, but not significantly different (200 ± 431 trials after crossing threshold compared to 500 ± 221 in BC; median ± SD, p > 0.05 singed rank test). Taken together, these results indicate that as mice gain expertise, prior to the sensation period, RL encodes history information, which may be later projected down onto BC where it is integrated with information of the current incoming texture.

## Past versus present discrimination power in BC and RL

How well can BC and RL activity discriminate at the single trial level past information compared to the information of the current stimulus? To do this, we computed the receiver operating characteristics (ROC) analysis between specific trial types (*Gilad et al., 2020*; *Gilad and Helmchen, 2020*), along with the area under the curve (AUC) quantifying the discrimination power at the single trial level (Materials and methods). We calculated the AUC between two types of trials (*Figure 6a*): (1) Activity between CR-Hit and Hit-Hit pairs based on the activity during the Hit trial. This is defined as History AUC since only the previous trial is different. (2) Activity between the current Hit and CR trials. This is defined as the current AUC because the current trial types are different (both in terms of stimulus type and action). Both history and current AUCs are calculated for BC and RL for each time frame along the trial structure and for naive, learning, and expert phases. Intuitively, one would assume that the current AUC will display higher discrimination power compared to the history AUC because the latter AUC measure compares the same previous trial type which should be harder to discriminate. Interestingly, during the expert phase, history AUC in both BC and RL has a discrimination power in the stim

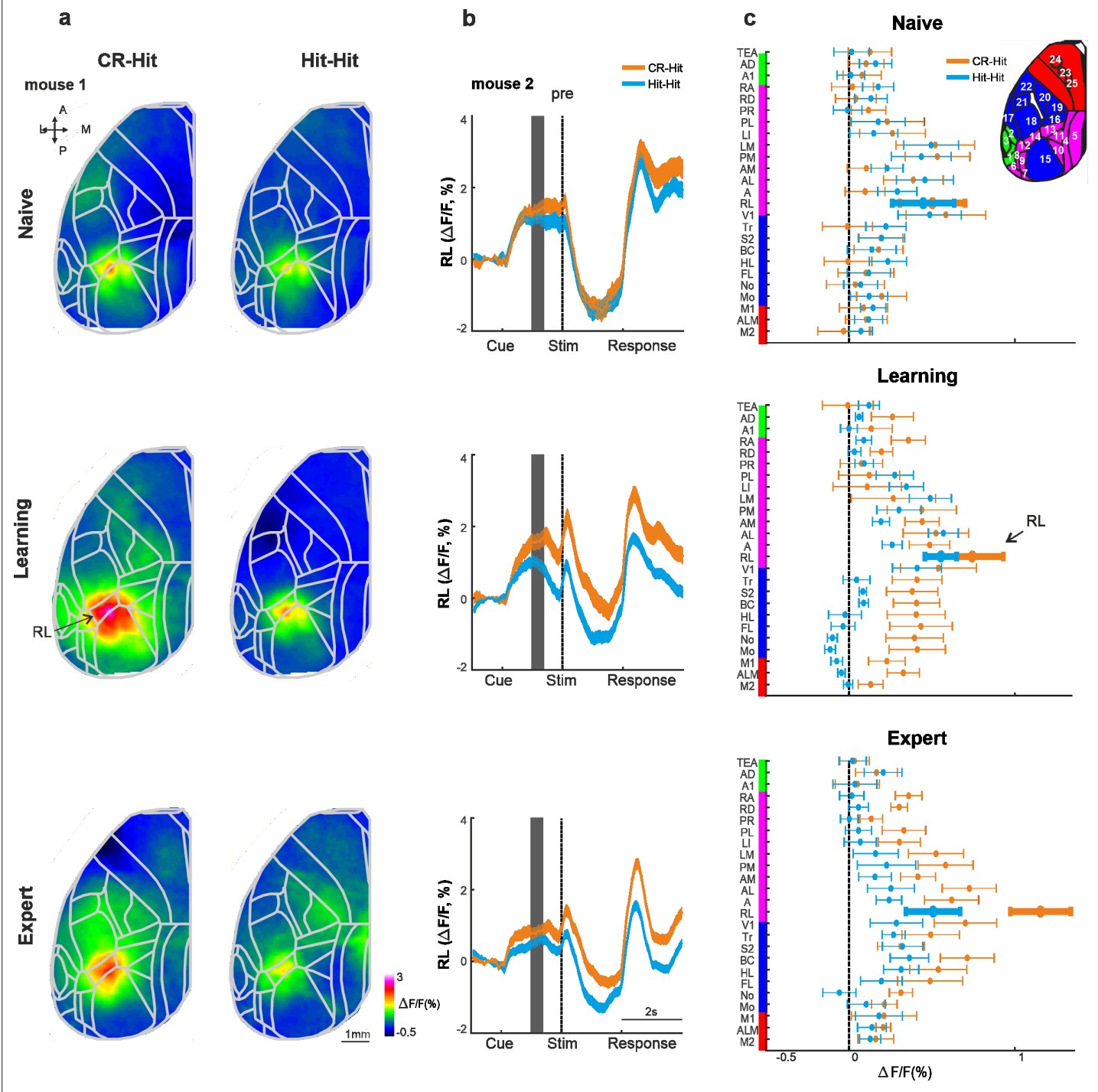

**Figure 5.** History information in rostrolateral (RL) before stimulus presentation. (**a**) Mean activity maps averaged within the pre period (−1 to −0.8 s relative to texture stop) of CR-Hit (left) Hit-Hit (right) during the naive (top), learning (middle), and expert (bottom) phases. Color bar denotes normalized fluorescence (ΔF/F). 2D top-view atlas is superimposed in gray. (**b**) Example from one mouse of average RL response of Hit-Hit (blue) and CR-Hit (orange) in the naive (top), learning (middle), and expert (bottom) phases. Shaded gray bar depicts the pre period (−1 to −0.6). Error bars are mean ± standard error of the mean (SEM) across trials (n = 51/54, 92/78, and 168/173 Hit-Hit/CR-Hit for naive, learning, and expert phases, respectively). (**c**) Grand average neuronal activity during the pre period (−1 to −0.6) for Hit-Hit (blue) and CR-Hit (orange) in all 25 areas for the naive (top), learning (middle), and expert (bottom) phases. Error bars depict mean ± SEM across mice (n = 7).

The online version of this article includes the following figure supplement(s) for figure 5:

**Figure supplement 1.** Activity in rostrolateral (RL) during the pre period.

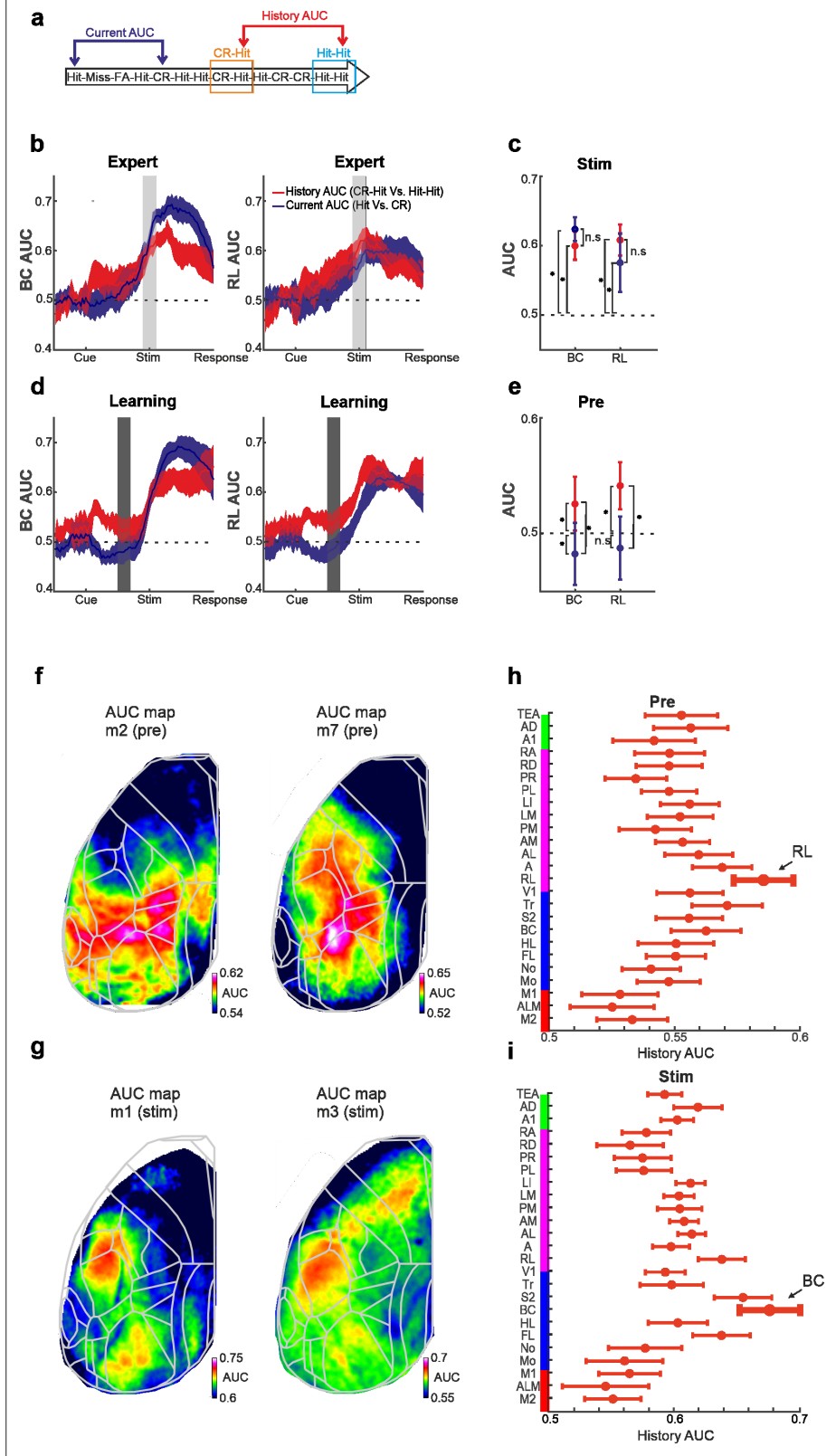

**Figure 6.** History and current information are equally discriminative at the single trial level. (**a**) Schematic diagram for the two types of area under the curve (AUC) measures (derived from a receiver operating characteristics [ROC] analysis): history AUC between the Hit responses for Hit-Hit and CR-Hit trial types. Current AUC between Hit and CR trial types regardless of their history. (**b**) Grand average of the history (red) and current (blue) AUC measures

*Figure 6 continued on next page*

*Figure 6 continued*

in barrel cortex (BC; left) and rostrolateral (RL; right) along the trial structure during the expert phase. Error bars depict mean ± standard error of the mean (SEM) across mice (*n* = 7 ). Values significantly differ from chance (0.5) in history AUC (p < 0.05, two-tailed *t*-test, for both BC and RL). (**c**) Grand average of history and current AUC measures during the stim period in the expert phase. Error bars as in **a**. (**d**) Same as in a but for the learning phase. Error bars as in a, values significantly differ from chance (0.5) for history AUC (p < 0.05, two-tailed *t*-test, for both BC and RL), but not for the current AUC in RL. (**e**) Same as in **c**, but for the pre period during the learning phase. *p < 0.05; n.s. – not significant; Wilcoxon signed-rank test. (**f**) History AUC map examples within the pre period. Each pixel in the map depicts the history AUC value, that is, the classifier accuracy between Hit-Hit and CR-Hit conditions. (**g**) Same as in f but average within the stim period. (**h**) History AUC values during the pre period for all the 25 cortical areas. Error bars depict number of mice (seven mice). (**i**) Same as in h, but for the stim period.

period that is not significantly different than that of the current AUC (*Figure 6b, c*; p > 0.05; singed rank test). In other words, we found that BC and RL discriminate past stimuli just as well as the current stimuli. In addition, during the learning phase, RL and to some extent BC, display a significantly higher history AUC compared to the current AUC, specifically in the pre period (*Figure 6d, e*; p < 0.05; singed rank test). This indicates that history information is discriminative at the single trial level before stimulus onset. Taken together, we find that BC and RL can encode the past just as well as the present.

We next calculated the history AUC for each pixel during either the pre or stim period (during expert phase). The history AUC maps during the pre period display AUC values around the RL areas (*Figure 6f*). In contrast, the history AUC maps during the stim period display AUC values mostly in BC (*Figure 6g*). Quantified across 25 areas and averaged across mice, RL displays the highest history AUC during the pre period, whereas BC displays the highest history AUC values during the stim period (*Figure 6h, i*). We note that additional cortical areas such as other association areas also display high history AUC values. In summary, we find that trial history emerges in RL before the texture arrives and then shifts to BC during stimulus presentation.

## Discussion

### History information is trial-type specific

We have identified cortex wide encoding of trial history information that emerges as mice learn to discriminate between two textures. Trial history encoding was not dependent on the current stimulus and emerged at RL association area before texture touch. Our results indicate that a previous CR trial will lead to higher activity in BC and RL compared to a previous Hit trial. This difference is probably not due to pure sensory differences in the previous trial since the effect was not present after FA trials (sup *Figure 2—figure supplement 7*, left panel). In addition, mice trained to lick the P1200 texture displayed a similar bias to the CR-Hit, further indicating that these differences are not purely sensory related. Moreover, this difference is probably not related to the previous motor action (e.g., either lick or no-lick). During the current trial, body and whisker movements were not significantly different, emphasizing that there are no motor-related differences based on the previous trial (*Figure 2c, d*). The fact that these differences emerged only after learning implies that these differences are not purely sensory or motor related but rather reflect internal history-related information. It may be that in a go/no-go discrimination task the mouse mainly learns not to lick for the no-go texture (i.e., CR), making the information of a CR trials more pronounced relatively to Hit trials. Another possibility is that a previous CR will cause a pronounced anticipatory state for the incoming texture, leading to enhanced cortical activity. Again, we did not find any consistent differences in motor movements based on the previous trials making this possibility less likely. In summary, our results indicate that history-dependent information emerges internally in cortex as mice learn to discriminate between two stimuli.

### Trial history emerges in RL and transferred to BC

BC is considered a lower-order sensory area but encodes not only lower-order stimulus features (*Chen et al., 2013*; *Estebanez et al., 2012*; *Garion et al., 2014*; *Safaai et al., 2013*) but also higher-order information such as choice and reward value (*Chéreau et al., 2020*; *Rodgers et al., 2021*; *Zuo and Diamond, 2019*). We additionally found that BC carries trial history information during the

sensation period which is related to the previous trial several seconds back. The presence of trial history information in lower-order areas such as BC is interesting by itself, but also raises the question of where is its origin. Interestingly, we show that trial history information emerges in RL before texture touch, implying that RL may transfer past experience in a top-down manner to BC for optimal sensory integration.

The presence of trial history in RL before the sensation period implies that RL may play a crucial role in linking past experience to ongoing sensory integration. RL is the lateral part of the PPC adjacent to BC, within the cluster of higher-order association areas surrounding V1 (*Hovde et al., 2019*; *Lyamzin and Benucci, 2019*). Previous studies showed that trial history of choice outcome is encoded by PPC neurons (*Harvey et al., 2012*; *Hwang et al., 2017*; *Morcos and Harvey, 2016*; *Pho et al., 2018*), as well as history of sensory information (*Akrami et al., 2018*). Silencing the PPC specifically during the inter-trial interval affected the behavioral performance of rats (*Akrami et al., 2018*; *Hwang et al., 2017*), whereas silencing during the stimulus presentation did not affected performance. The PPC is also reciprocally connected to hippocampus via entorhinal and retrosplenial cortices (*Save and Poucet, 2009*; *Whitlock et al., 2008*) and to basolateral amygdala via the anterior cingulate cortex (*Suzuki et al., 2022*), giving fast access to the different memory hubs. (*Khodagholy et al., 2017*) showed coupling of PPC and hippocampal ripples that strengthen in non-REM (Rapid eye movement) sleep after rats learned a spatial exploration task, further indicating that RL may relay history information from subcortical memory hubs to cortex.

The fact that trial history emerges only after learning, implies that it encodes a subjective value or association of a certain past stimulus. It may be that only once the value of a certain stimulus is established, for example, by strengthening indirect connections between basolateral amygdala (that has a role in associative memory) and RL, history information can aid in efficiently encoding the incoming stimulus. In light of this discussion, we suggest that the consolidation of a certain association (in our case a CR), induces long-term synaptic plasticity of top-down projections from higher-order association area (e.g., RL) to a lower-order sensory area (e.g., BC). This projection-specific potentiation may facilitate the recruitment sensory cortex in the context of the immediate previous history.

## Mechanisms for integrating past and present

The wide-field signal measured in our study reports bulk population activity specifically in L2/3 excitatory cells. Are neuronal populations encoding past and present information in the BC overlapping or distinct? On the one side, it could be that the same cell in BC encodes both the current stimulus and additionally receives top-down input from RL carrying the past stimulus identity. This additional top-down information may amplify sensory integration and optimize discrimination of the current stimulus. On the other side, previous studies that measured single cell activity in the BC showed that single cells tend to respond to one information type (*Chéreau et al., 2020*; *Estebanez et al., 2012*; *Rodgers et al., 2021*). In this case, we hypothesize that different populations in BC encode current and history information, which leads to a larger fraction of neurons in BC that are active for the CR-Hit pair. A larger number of active neurons in BC may facilitate sensorimotor integration involving downstream areas such as the motor cortex, further resulting in gaining expertise (*Zuo and Diamond, 2019*).

It is probable that both history and learning involve other circuit elements such as deep cortical layers (*Pasupathy and Miller, 2005*; *Roelfsema and Holtmaat, 2018*; *Vecchia et al., 2020*), inhibitory subtypes, other pathways (*Lacefield et al., 2019*; *Mohan et al., 2022*; *Musall et al., 2023*; *Petreanu et al., 2012*; *Williams and Holtmaat, 2019*), and subcortical areas (*Fu et al., 2015*; *Garrett et al., 2020*; *Pasupathy and Miller, 2005*; *Pfeffer et al., 2013*). Future work may aim to dissect specific subpopulations that carry history information using similar behavioral tasks, for example, imaging of cortex-wide layer 5 dynamics. Layer 5 neurons may be ideal in integrating past information arriving onto the apical dendrites in layer 1 (*Petreanu et al., 2012*) with incoming information arriving from the thalamus. In addition, similar task with reward after CR trails, or tasks that better differentiate between choice and outcome (decision tasks, giving different probabilities of outcome to each choice), or tasks with a dynamic inter-trial interval may shed light on the meaning of this history-learning effect. In summary, our results imply that as we learn, the cortex learns to better integrate past and present information resulting in expert performance.

## Materials and methods

### Animals and surgical procedures

Methods were carried out according to the guidelines of the Veterinary Office of Switzerland and following approval by the Cantonal Veterinary Office in Zurich and by the Institutional Animal Care and Use Committee (IACUC) at the Hebrew University of Jerusalem, Israel (Permit Number: MD-20-16065-4). A total of seven adult male mice (1–4 months old) were used in this study. These mice were triple transgenic Rasgrf2-2A-dCre; CamK2a-tTA;TITL-GCaMP6f animals, expressing GCaMP6f in excitatory neocortical layer 2/3 neurons (*Gilad and Helmchen, 2020*). The dataset used here is identical to our previous study (*Gilad and Helmchen, 2020*), but here we have applied a completely novel history analysis. To generate triple transgenic animals, double transgenic mice carrying CamK2a-Tta62 and TITL-GCaMP6f63 were crossed with a Rasgrf2-2A-dCre line (64; individual lines are available from The Jackson Laboratory as JAX# 016198, JAX#024103, and JAX# 22864, respectively). The Rasgrf2-2A-dCre;CamK2a-tTA;TITL-GCaMP6f line contains a tet-off system, by which transgene expression can be suppressed upon doxycycline treatment (*Garner et al., 2012*; *Gossen and Bujard, 1992*). However, doxycycline treatment is not necessary in these animals, since the Rasgrf2-2A-dCre line holds an inducible system of its own, given that the destabilized Cre (dCre) expressed under the control of the Rasgrf2-2A promoter needs to be stabilized by trimethoprim (TMP) to be fully functional. TMP (Sigma T7883) was reconstituted in Dimethyl sulfoxide (DMSO, Sigma 34869) at a saturation level of 100 mg/ml, freshly prepared for each experiment. For TMP induction, mice were given a single intraperitoneal injection (150 µg TMP/g body weight; 29 g needle; 3–5 days post-surgery), diluted in 0.9% saline solution. We used an intact skull preparation (*Silasi et al., 2016*) for chronic wide-field calcium imaging of neocortical activity (*Gilad et al., 2018*). Mice were anesthetized with 2% isoflurane (in pure O₂) and body temperature was maintained at 37°C. We applied local analgesia (lidocaine 1%), exposed and cleaned the skull, and removed some muscles to access the entire dorsal surface of the left hemisphere (*Figure 2a*; ~6 × 8 mm² from ~3 mm anterior to bregma to ~1 mm posterior to lambda; from the midline to at least 5 mm laterally). We built a wall around the hemisphere with adhesive material (iBond; UV-cured) and dental cement 'worms' (Charisma). Then, we applied transparent dental cement homogenously over the imaging field (Tetric EvoFlow T1). Finally, a metal post for head fixation was glued on the back of the right hemisphere. This minimally invasive preparation enabled high-quality chronic imaging with high success rate.

### Texture discrimination task

Mice were trained on a go/no-go discrimination task (*Figure 1a*) using a data acquisition interface (USB-6008; National Instruments) and custom-written LabVIEW software (National Instruments) available as a source code file (*Gilad, 2016*). Each trial started with an auditory cue (stimulus cue; 2 beeps at 2 kHz, 100 ms duration with 50 ms interval), signaling the approach of either two types of sandpapers (grit size P100: rough texture; P1200: smooth texture; 3M) to the mouse's whiskers as 'go' or 'no-go' textures (*Figure 1a*; pseudo-randomly presented with no more than three repetitions). Sandpapers were mounted onto panels attached to a stepper motor (T-NM17A04; Zaber) mounted onto a motorized linear stage (T-LSM100A; Zaber) to move textures in and out of reach of whiskers. The texture stayed in touch with the whiskers for 2 s, and then it was moved out after which an additional auditory cue (response cue; 4 beeps at 4 kHz, 50 ms duration with 25 ms interval) signaled the start of a 2-s response period. The stimulus and response cues were identical in both textures. The interval between the trails was 6 s (8 s from response to next cue). A water reward (~3 µl) was given to the mouse for licking for the go texture only after the response cue ('Hit'), that is for the first correct lick during the response period (*Figure 1a*; lick were detected using a piezo sensor). Punishment with white noise was given for licking for the no-go texture ('false alarms'; FA). Licking before the response cue was neither rewarded nor punished. Reward and punishment were omitted when mice withheld licking for the no-go ('correct-rejections', CR) or go ('Misses') textures.

### Training and performance

Five mice were trained to lick for the P100 texture (mice #1–4 and 6) and two mice were trained to lick for the P1200 texture (mice #5 and 7). Mice were first handled and accustomed to head fixation before starting water scheduling. Before imaging began mice were conditioned to lick for reward after the go texture (presented within a similar trial structure as the task itself). Imaging began only after

mice reliably licked for the response cue (typically after the first day; 200–400 trials). On the first day of imaging, mice were presented with the 'go' texture and after 50 trials the 'no-go' texture was gradually introduced (starting from 10% and increasing by 10% approximately every 50 trials; *Guo et al., 2014*) until reaching 50% probability for the no-go texture by the end of the day. Six out of the seven mice learned the task within 3–4 days after around a thousand trials (Supplementary Fig. 1). Mouse #4 learned the task within 10 days. An effort was made to maintain a constant position of the texture and cameras across imaging days in order to maintain similar stimulation and imaging parameters.

## Wide-field calcium imaging

We used a wide-field approach to image large parts of the dorsal cortex while mice learned to perform the task (*Gilad et al., 2018*). A sensitive CMOS camera (Hamamatsu Orca Flash 4.0) was mounted on top of a dual objective setup. Two objectives (Navitar; top objective: D-5095, 50 mm f0.95; bottom objective inverted: D-2595, 25 mm f0.95) were interfaced with a dichroic (510 nm; AHF; Beamsplitter T510LPXRXT) filter cube (Thorlabs). This combination allowed a ~9 mm field-of-view, covering most of the dorsal cortex of the hemisphere contralateral to texture presentation. Blue LED light (Thorlabs; M470L3) was guided through an excitation filter (480/40 nm BrightLine HC), a diffuser, collimated, reflected from the dichroic mirror, and focused through the bottom objective ~100 µm below the blood vessels. Green light emitted from the preparation passed through both objectives and an emission filter (514/30 nm BrightLine HC) before reaching the camera. The total power of blue light on the preparation was <5 mW; that is, <0.1 mW/mm$^2$. At this illumination power we did not observe any photobleaching. Data were collected with a temporal resolution of 20 Hz and a spatial sampling of 512 × 512 pixels, resulting in a spatial resolution of ~20 µm/pixel. On each imaging day a green reflectance image was taken as reference to enable registration across different imaging days using the blood vessel pattern (fibercoupled LED illuminated from the side; Thorlabs).

## Mapping and area selection

Each mouse underwent a mapping session under anesthesia (1% isoflurane), in which we presented five different sensory stimuli (contra-lateral side) (*Garion et al., 2014*). Next, we registered each imaging day to the mapping day using skull coordinates from the green images. Finally, we registered each mouse onto a 2D top view mouse atlas using both functional patches from the mapping and skull coordinates (*Garion et al., 2014*; 2004 Allen Institute for Brain Science. Allen Mouse Brain Atlas. Available from http://mouse.brain-map.org/29). Within the atlas borders, we defined 25 areas of interest, with some manual modifications within these borders to fit the functional activity for each mouse. Motor cortex areas were defined based on stereotaxic coordinates and functional patches for each mouse (see below). Thus, all mice had similar regions of interest that were comparable within and across mice. We grouped these 25 areas into auditory (green), association (pink), somatosensory + V1 (blue), and motor (red) areas (*Figure 1d* ). Auditory areas: primary auditory (A1), auditory dorsal (AD), and temporal association area (TEA). Sensory areas: somatosensory mouth (Mo), somatosensory nose (No), somtosensory hindlimb (HL), somtosensory forelimb (FL), barrel cortex (BC; primary somatosensory whisker); secondary somatosensory whisker (S2), somtosensory trunk (Tr), and primary visual cortex (V1). Motor areas: whisker-related primary motor cortex (M1; 1.5 anterior and 1 mm lateral from bregma, corresponding to the whisker evoked activation patch in M1 from the mapping session), anterior lateral motor cortex (ALM; 2.5 anterior and 1.5 mm lateral from bregma) and secondary motor cortex (M2; 1.5 anterior and 0.5 mm lateral from bregma corresponding; *Gilad et al., 2018*). Association cortex: rostrolateral (RL), anterior (A), anterior lateral (AL), anterior medial (AM), posterior medial (PM), lateral medial (LM), lateral intermediate (LI), posterior lateral (PL), post-rhinal (PR), retrosplenial dorsal (RD), and retrosplenial angular (RA). We note that our definition of association cortex is broad and may include or exclude areas that are not necessarily classical association areas.

## Control for non-calcium-dependent signals

The data collected in this study used a single wavelength (473 nm) to image calcium dynamics (similar to *Gallero-Salas et al., 2021*; *Gilad et al., 2018*; *Gilad and Helmchen, 2020*). This protocol may additionally collect non-calcium-dependent signal, such as hemodynamic signal, which may affect the results. To control for this, we performed several steps:

1. In two out of the seven original mice we also imaged expert mice with a block session using an isosbestic control light (510 nm). Responses in cortex and specifically BC were relatively low, displaying a gradual decrease after texture stop (*Figure 2—figure supplement 2a*). By correcting an adjacent 473-light session based on the 510 nm session (473 signal minus 510 signal), trial history was maintained, that is, responses in BC were significantly higher in CR-Hit compared to Hit-Hit (*Figure 2—figure supplement 2a*).

2. We replicated the experiment in three additional mice using an interleaved imaging protocol of 473 and 405 nm (isosbestic) excitation lights (*Figure 2—figure supplement 3*; 10 Hz for each signal; Using a teensy 3.5 for light alteration). Mice further underwent a craniotomy to implant a 5-mm window covering most of the posterior cortex (*Figure 2—figure supplement 3b*). Correcting for non-calcium signals (473 light minus 405 light within each trial) maintain trial history finding, that is, we find a significant difference between CR-Hit and Hit-Hit in RL and BC during the pre and stim periods, respectively. This was true during learning and expert phases but not during the naive phase (*Figure 2—figure supplement 3d*)

3. The three mice then continued to two-photon imaging of single cells in BC and RL. Two-photon imaging single cell imaging is less prone to hemodynamic artifacts. Localization of BC and RL was done by aligning functional patches and blood vessel patterns obtained from the same mice in the wide-field system. We used a mesoscope (Thorlabs) and imaged each area (separately or simultaneously) with a temporal resolution of 44.7 Hz (or 22.8 Hz for simultaneous imaging). Data were collected, and went through a preprocessing pipeline that included background subtraction, X–Y movement correction (based on frame-to-frame optimal correlation correction), manual cell body selection, and frame-zero division (20 frames before cue onset similar to the wide-field signal). Single cells were screened for responsiveness by exceeding a 2 STD activity during pre and stim periods as compared to baseline. Next, single cell responses were divided to CR-Hit and Hit-Hit pairs similar to the wide-field signals. In general, we found significant differences in single cell activity (i.e., higher response in CR-Hit compared to Hit-Hit) in RL and BC during the pre and stim phases, respectively (*Figure 2—figure supplement 4*)

Taken together, we were able to replicate our finding by either controlling for non-calcium contamination or directly imaging single cells in BC and RL.

## Whisker and body tracking

In addition to wide-field imaging, we tracked movements of the whiskers and the body of the mouse during the task (*Figure 1a*). The mouse was illuminated with a 940-nm infrared LED. Whiskers were imaged at 50 Hz (500 × 500 pixels) using a high-speed CMOS camera (A504k; Basler), from which we calculated time course of whisking envelope and the time of first touch (see below). An additional camera monitored the movements of the mouse at 30 Hz (The imaging source; DMK 22BUC03; 720 ×48 0 pixels). We used movements of both forelimbs and the head/neck region to assess body movements, to reliably detect large movements (*Figure 1a*; see Data analysis).

## Calculating body movements

We used a body camera to detect general movements of the mouse (30 Hz frame rate). For each imaging day, we first outlined the forelimbs and the neck areas (one area of interest for each), which were reliable areas to detect general movements. Next, we calculated the body movement (1 minus frame-to-frame correlation) within these areas as a function of time for each trial. We than averaged all the defined body areas to one 'body' vector. As a more detailed analysis, we tracked 22 individual body points using DeepLabCut (*Mathis et al., 2018*; *Figure 2—figure supplement 5a*; *Mathis et al., 2018*). For each tracking point we calculated the Euclidian distance between consecutive frames and compared trial history during naive, learning, and expert phases.

## Whisker tracking

The average whisker angle across all imaged whiskers was measured using automated whisker tracking software (*Knutsen et al., 2005*). The mean whisker envelope was calculated as the difference between maximum and minimum whisker angles along a sliding window equal to the imaging frame duration (50 ms; *Gilad et al., 2018*). Whisker envelope was normalized just before the auditory cue similar to wide-field data (Frame zero). In a more detailed analysis, we tracked single whiskers using DeepLabCut (*Mathis et al., 2018*) and calculated single whisker kinematics (*Figure 2—figure supplement 6*). Single whisker parameters were compared between CR-Hit and Hit-Hit conditions

(*Figure 2—figure supplement 6*). In addition, we manually detected the first frame, in which any whisker touched the upcoming texture, using the movies from the whisker camera (LabVIEW custom program). The first touch occurred on average 0.33 and 0.34 s before the texture stopped for naive and expert mice, respectively. Time of first touch did not differ between expert and naive mice (p > 0.05; Mann–Whitney *U*-test; *n* = 7 mice). We note that the first touch occurred mostly (but not exclusively) in the pre period from −1 to −0.5 relative to texture stop.

## Data analysis

Data analysis was performed using Matlab software (Mathworks). All mice were continuously imaged during learning (5–11 days). Wide-field fluorescence images were sampled down to 256 × 256 pixels and pixels outside the imaging area were discarded. This resulted in a spatial resolution of ~40 μm/pixel and was sufficient to determine cortical borders, despite further scattering of emitted light through the tissue and skull. Each pixel and each trial were normalized to baseline several frames before the stimulus cue (frame 0 division). Our main focus was on the history effect. Because the hit trails had the largest portion from all trails, we focused on the hit trials. We subgrouped all the Hit trials based on the type of the preceding trial as follows: CR-Hit – Hit trials that were preceded by CR trial. Hit-Hit – Hit trials that were preceded by a Hit trial. FA-Hit – hit trials that were preceded by an FA trial. We mainly focused on comparing Hit-Hit and CR-Hit pairs since they had a large proportion in naive, learning, and expert phases (but see *Figure 2—figure supplement 7*). We defined two time periods within the trial structure: pre (−1 to 0.6 s relative to texture stop) and stim (−0.2 to 0.2 relative to texture stop; *Figure 1a*).

## Calculation of learning curves

Trials were binned (*n* = 100 trials with no overlap) across learning (at the stimulus time, adjusted for each mouse) and the performance (defined as *d′* = *Z*(Hit/(Hit + Miss)) − *Z*(FA/(FA + CR)), where *Z* denotes the inverse of the cumulative distribution function) was calculated for each bin. Next, each behavioral learning curve was fitted with a sigmoid function $s(t) = a \frac{1}{1+e^{\frac{-(t-b)}{c}}}$, where *a* denotes the amplitude, *b* the time point (in trial numbers) of the inflection point, and *c* the steepness of the sigmoid.

A learning threshold was defined as the bin in which the *d′* crossed the inflection point (half point) of the learning curve sigmoid fit (*Figure 1—figure supplement 1*).

## Defining the learning phases

We defined the naive, learning, and expert phase each as 1 day of recordings, the naive day was defined as the first day to have enough CRs that the performance is still before the crossing threshold (typically the second recording day). The learning day was defined as the day that the mouse crossed the learning threshold, and the expert was defined as the last day of the mouse (usually the fifth day).

## Calculating history modulation and onset

We defined the 'history modulation' as the difference between the average activation of all CR-Hit and Hit-Hit trials. To calculate significance of history modulation, we calculated the sample distribution by trial shuffling between CR-Hit and Hit-Hit trials (*n* = 1000 iterations). We than defined the onset of the history modulation as the first bin exceeding mean ± 2 SD of the sample distribution. We calculated this history modulation and significance across the trial dimension (every frame) and across learning dimension (every 100 trials). In the learning dimension, we calculated the average activity in the stim period (−0.2:0.2) of all the CR-Hit and Hit-Hit trials that were falling within each 100 trials bin.

## Discrimination power between hit trials subgrouped by history

To measure how well could neuronal populations discriminate between go and no-go textures, we calculated an ROC curve and calculated its AUC (with a value of 0.5 indicating no discrimination power). This can be done for a given area, each time frame within each learning phase separately (*Figure 6*).

## Statistical analysis

In general, the Wilcoxon signed-rank test was used to compare a population's median to zero (or between two paired populations). For non-paired populations we used a Mann–Whitney *U*-test

to compare between medians. A two-way repeated measure ANOVA was used to relate between learning and history in BC and RL separately. Multiple group correction was used when comparing between more than two groups.

## Acknowledgements

This project has received funding from the European Union's Horizon 2020 research and innovation program under the Marie Skłodowska-Curie grant agreement No. 659719 (AG). This work was supported by grants from Hebrew University of Jerusalem (Start-up grant; AG), the European Union ERC starting grant, MESO-AG, No. 101040378 (AG), and the Swiss National Science Foundation (SNSF) (31003A-149858 and 310030B_170269; FH).

## Additional information

### Funding

| Funder | Grant reference number | Author |
| --- | --- | --- |
| Marie Curie - Global Fellowship | 659719 | Ariel Gilad |
| Hebrew University of Jerusalem, Start up grant | | Ariel Gilad |
| Swiss National Science Foundation | 31003A-149858 | Fritjof Helmchen |
| Swiss National Science Foundation | 310030B_170269 | Fritjof Helmchen |
| European Union ERC starting grant, MESO-AG | 101040378 | Ariel Gilad |

The funders had no role in study design, data collection, and interpretation, or the decision to submit the work for publication.

### Author contributions

Odeya Marmor, Conceptualization, Software, Formal analysis, Visualization, Writing - original draft; Yael Pollak, Data curation, conducted additional control experiments; Chen Doron, Data curation, conducted additional control experiments; Fritjof Helmchen, Resources, Funding acquisition, Methodology, Writing - review and editing; Ariel Gilad, Conceptualization, Data curation, Formal analysis, Supervision, Funding acquisition, Investigation, Visualization, Methodology, Writing - original draft, Writing - review and editing

### Author ORCIDs

Odeya Marmor ⬚ http://orcid.org/0000-0002-0290-5888
Fritjof Helmchen ⬚ http://orcid.org/0000-0002-8867-9569
Ariel Gilad ⬚ http://orcid.org/0000-0001-8802-8611

### Ethics

Methods were carried out according to the guidelines of the Veterinary Office of Switzerland and following approval by the Cantonal Veterinary Office in Zurich (Permit Number: MD-20-16065-4).

### Decision letter and Author response

Decision letter https://doi.org/10.7554/eLife.83702.sa1
Author response https://doi.org/10.7554/eLife.83702.sa2

## Additional files

### Supplementary files
• MDAR checklist

• Source code 1. Behavioral GoNoGo program.

## Data availability

The data and custom code that support the findings of this study are publicly available at https://doi.org/10.17605/OSF.IO/HKVC5.

The following previously published dataset was used:

| Author(s) | Year | Dataset title | Dataset URL | Database and Identifier |
|---|---|---|---|---|
| Marmor O | 2022 | HistoryLearning | https://doi.org/10.17605/OSF.IO/HKVC5 | Open Science Framework, 10.17605/OSF.IO/HKVC5 |

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
