## [Editor Report]

This is important work analyzing the trial-by-trial progression of learning, and how the outcome of one trial influences cortex-wide neural responses on the next trial. The strength of the evidence is compelling, with control experiments provided to rule out potential confounds of hemodynamic effects and extensive analyses provided to address the challenging issue of potential behavioral changes induced by the previous trial.

---

## [Decision Letter]

**Decision letter after peer review:**

[Editors’ note: the authors submitted for reconsideration following the decision after peer review. What follows is the decision letter after the first round of review.]

Thank you for submitting the paper "History information emerges in the cortex during learning" for consideration by *eLife*. Your article has been reviewed by 2 peer reviewers, and the evaluation has been overseen by a Reviewing Editor and a Senior Editor. The reviewers have opted to remain anonymous.

Comments to the Authors:

We are sorry to say that, after consultation with the reviewers, we have decided that this work will not be considered further for publication by *eLife*.

The analysis of trial history on neural responses across the cortex, and how they emerge over the course of learning is a scientific area of broad interest. However several potential confounds call into question the central finding about the effect of the preceding trial on neural responses during the next trial :

1) Slow timescale hemodynamics can potentially explain the Hit-Hit/CR-Hit difference and need to be controlled for.

2) Body movements may well differ on rewarded and unrewarded trials can also potentially account for the Hit-Hit/Cr-Hit difference and need to be controlled for (and unfortunately this can drive hemodynamic differences that could account for the result – for instance, if whisker posture is different after a rewarded and unrewarded trial, which is very common, one could imagine sustained greater blood flow to BC in one case and not the other).

3) Statistical analyses need improvement – both because a small sample size makes insignificant outcomes highly likely even when in fact things are significant, and because in several places two sample tests are inappropriate.

*Reviewer #1 (Recommendations for the authors):*

1) I found the second paragraph of the introduction a bit hard to follow, especially with regard to what the outstanding questions in the field are. For instance, what does it mean to "link history information with the learning process"? What about the "temporal aspect that enables integration of past information with present sensory information"?

2) Line 57. Typo. "Does history information emerges …?" (Should be emerge).

*Reviewer #2 (Recommendations for the authors):*

1. Whisker plots with n=7 seem a bit excessive; perhaps just show the raw data points and a central tendency? (Figures S3, S4).

2. In several cases, points seem to be superimposed – if points overlap, as seems to be the case in Figure 2d, please separate them horizontally so that it does not appear that data is missing. If n=5, you will never get p <.05 in which case n.s. in 2d is trivial.

3. Perhaps use the term "trial history"/"sensory history"/"reward history" instead of history information – "history information" is less precise.

---

## [Author Response]

[Editors’ note: The authors appealed the original decision. What follows is the authors’ response to the first round of review.]

Comments to the Authors:We are sorry to say that, after consultation with the reviewers, we have decided that this work will not be considered further for publication by eLife.The analysis of trial history on neural responses across the cortex, and how they emerge over the course of learning is a scientific area of broad interest. However several potential confounds call into question the central finding about the effect of the preceding trial on neural responses during the next trial :

We have taken the reviewers' concern very seriously and performed additional experiments and controls to show that the central findings regarding history information are completely maintained. Basically, we have redone the whole study. Please see details below

1) Slow timescale hemodynamics can potentially explain the Hit-Hit/CR-Hit difference and need to be controlled for.

This is now completely addressed. First, we show control results from 2 mice that are in the original data set (m3 and m6). For these mice, we used also a non-calcium isosbestic light (510 nm; Mainly targeting hemodynamic signals) as mice performed the task (expert case). It is clear that the signal obtained from the control light is relatively weak with a typical slow decrease around stimulus onset (Figure S4 shown here). Importantly, when correcting the calcium signal (473 nm) to the hemodynamic signal, trial history information differences are significantly maintained in BC. In general, we have to mention that this mouse line (triple transgenic Layer 2/3 excitatory cells) has a much stronger calcium SNR compared to the hemodynamic intrinsic signal and we have already published 3 papers (2 in Neuron and one in Nat Comm.) with this type of imaging protocol (i.e., only 473 light).

Second, to thoroughly address this concern, we have basically redone the experiments in 3 mice while controlling for potential effects of non-calcium signal on the results. In short, we trained three mice on the same task and imaged throughout learning using an interleaved protocol of alternating calcium (473 nm) and control (405 nm) lights. This enabled us to correct for hemodynamic signals on a trial-wise basis. In short, results were maintained in both BC and RL (Figure S5 below; We also present within mouse statistics for the 3 mice).

Third, we performed additional and novel experiments using two-photon microscopy which is not affected by hemodynamic artifacts (Figure S6). In short, we find similar differences between CR-Hit and Hit-Hit conditions using a single cell approach.

In summary, we have performed additional experiments showing that hemodynamic modulations do not affect the results, specifically trial history information.

We have now added three Supplementary figures and revised the Results section which now reads (pg. 4): "To control for the possible contamination of non-calcium related signals such as hemodynamics, we performed a battery of additional experiments. First, two mice performed the task (expert) while we excited the cortex with a control light (510 nm, isosbestic wavelength). Correcting the original signal with the control light maintained significant difference in trial history (Figure S4). Second, we trained an additional 3 mice on the same task and imaged their cortex using an interleaved protocol of 473 (calcium signal) and 405 (control signal, isosbestic wavelength) nm lights. The corrected signal (473 signal minus the 405 signal; See Methods) maintained a significant trial history difference between CR-Hit and Hit-Hit conditions during learning and expert phases, but not during the naïve phase (Figure S5). Finally, we further performed two photon imaging of single cells in BC and RL during the expert phase and found significant trial history differences, i.e., higher response in CR-Hit compared to Hit-Hit in BC (during the stim period) and RL (Figure S6; during the pre period). Taken together, noncalcium dynamics such as hemodynamics have a minor effect on the results, specifically regarding trial history differences."

2) Body movements may well differ on rewarded and unrewarded trials can also potentially account for the Hit-Hit/Cr-Hit difference and need to be controlled for (and unfortunately this can drive hemodynamic differences that could account for the result – for instance, if whisker posture is different after a rewarded and unrewarded trial, which is very common, one could imagine sustained greater blood flow to BC in one case and not the other).

Once resolving the hemodynamic issue raised in point 1, it might still be possible that some changes in movement or whisker position (even if it happened 10 seconds before) could have some effect on history information. I am one of the pioneers in addressing body movement in relation to wide field signals as can be seen in Gilad et al. 2018 and 2020a, b and Gallero-Salas et al. 2021. For this reason, we give a detailed movement analysis already in Figure 2c, d. From the example in Figure 2c, especially in the expert case, it is evident there are almost no differences in body movements or whisker envelope.

Nevertheless, we took this concern very seriously. We now meticulously examined and analyzed body and whisker parameters showing that there is no significant difference in body parameters between Hit-Hit and CR-Hit conditions. Please see relevant points below for more details.

3) Statistical analyses need improvement – both because a small sample size makes insignificant outcomes highly likely even when in fact things are significant, and because in several places two sample tests are inappropriate.

We have now added additional statistical analyses beyond the original draft and beyond the statistics presented in Gilad and Helmchen 2020. We also add within mouse statistics per each original mouse (n=7) and also of three additional mice (Figure S5). Importantly, the additional statistics further substantiate the results. Please see detailed response below.

Reviewer #1 (Recommendations for the authors):1) I found the second paragraph of the introduction a bit hard to follow, especially with regard to what the outstanding questions in the field are. For instance, what does it mean to "link history information with the learning process"? What about the "temporal aspect that enables integration of past information with present sensory information"?

Done. Paragraph 2 in the introduction is now revised accordingly.

2) Line 57. Typo. "Does history information emerges …?" (Should be emerge).

Fixed.

Reviewer #2 (Recommendations for the authors):1. Whisker plots with n=7 seem a bit excessive; perhaps just show the raw data points and a central tendency? (Figures S3, S4).2. In several cases, points seem to be superimposed – if points overlap, as seems to be the case in Figure 2d, please separate them horizontally so that it does not appear that data is missing. If n=5, you will never get p <.05 in which case n.s. in 2d is trivial.

Done. Superimposed points are now separated across MS.

3. Perhaps use the term "trial history"/"sensory history"/"reward history" instead of history information – "history information" is less precise.

Done. We have now reverted to the term 'trial history' throughout the MS.